# Neuroimaging of Mouse Models of Alzheimer’s Disease

**DOI:** 10.3390/biomedicines10020305

**Published:** 2022-01-28

**Authors:** Amandine Jullienne, Michelle V. Trinh, Andre Obenaus

**Affiliations:** Department of Pediatrics, University of California, Irvine, CA 92697, USA; ajullien@uci.edu (A.J.); mvtrinh@uci.edu (M.V.T.)

**Keywords:** Alzheimer’s Disease, mouse models, magnetic resonance imaging, positron emission tomography, 5xFAD, 3xTg-AD

## Abstract

Magnetic resonance imaging (MRI) and positron emission tomography (PET) have made great strides in the diagnosis and our understanding of Alzheimer’s Disease (AD). Despite the knowledge gained from human studies, mouse models have and continue to play an important role in deciphering the cellular and molecular evolution of AD. MRI and PET are now being increasingly used to investigate neuroimaging features in mouse models and provide the basis for rapid translation to the clinical setting. Here, we provide an overview of the human MRI and PET imaging landscape as a prelude to an in-depth review of preclinical imaging in mice. A broad range of mouse models recapitulate certain aspects of the human AD, but no single model simulates the human disease spectrum. We focused on the two of the most popular mouse models, the 3xTg-AD and the 5xFAD models, and we summarized all known published MRI and PET imaging data, including contrasting findings. The goal of this review is to provide the reader with broad framework to guide future studies in existing and future mouse models of AD. We also highlight aspects of MRI and PET imaging that could be improved to increase rigor and reproducibility in future imaging studies.

## 1. Introduction

Alzheimer’s Disease (AD) is the most common type of dementia, and in the United States is the fifth leading cause of death in adults aged 65 or older [1]. In 2021, it has been estimated that 6.2 million Americans 65 or older live with AD [2], and this number is expected to reach 14 million by 2060 [3]. Recent estimates of dementia-related mortality suggest that survival after a diagnosis ranges from 3 to 6 years [4]. A variety of risk factors contribute to AD, including the presence of one or more alleles of APOE ε4, advancing age, cardiovascular factors, and the presence of additional genes (e.g., TREM2). Recently, polygenic risk scoring has come to the fore to facilitate diagnosis, as well as providing potential targets for further research [5,6].

AD has been classified into two categories: early (prior to age 65; EOAD) and late (after age 65; LOAD) onset. The majority of AD cases (~95%) are sporadic AD, while the remaining ~5% of AD represent early onset familial (fAD) [7]. Mutations in the amyloid precursor protein (APP) and presenilin (PSEN1 and PSEN2) genes are important contributors to early onset familial variants [8]. LOAD is a continuum of symptoms that progress from mild cognitive impairment (MCI) to severe dementia, often spanning decades. Clinical signs of AD include progressive memory loss and decline in cognitive function [2]. While the exact biological mechanisms of pathogenesis are unclear in LOAD, cellular hallmarks include the extracellular accumulation of beta-amyloid (Aβ) peptides into senile plaques, the intracellular accumulation of hyperphosphorylated tau into neurofibrillary tangles (NFTs), glial (astrocytic and microglial) responses, and neuronal and synaptic loss [9,10]. However, LOAD cases are multifactorial and often are associated with multiple pathologies, including vascular and metabolic alterations, and thus identifying contributors is difficult. In 2018, the National Institute on Aging and the Alzheimer’s Association formed a working group to provide revised guidelines, which resulted in the AT(N) biomarker designations [11]. In AT(N), A stands for biomarkers of Aβ plaques, T represents fibrillar tau, and biomarkers of neurodegeneration are represented by (N). Each biomarker group has both an imaging biomarker and a cerebrospinal fluid (CSF) reporter for each component of the AT(N) scheme. CSF profiles can be used across all three domains, but positive positron emission tomography (PET) imaging is needed for Aβ and tau confirmation, while magnetic resonance imaging (MRI) can be used as a neurodegeneration marker surrogate. Importantly, the AT(N) measures allow for the definition of AD as well as the staging (or biomarker profile) of disease severity. For additional details, Ferrari and Sorbi have eloquently reviewed the complexity of AD [12].

A definitive diagnosis of AD can only be achieved post-mortem by linking clinical symptoms to the abnormal presence of senile plaques and/or NFTs in brain tissue. Neuroimaging techniques including MRI and PET have dramatically advanced our ability to identify putative AD patients (see below). Significant efforts in both blood- and CSF-based biomarkers have started to show promise, as several reviews summarize [13,14,15].

Substantial efforts have been put forth to identify or develop appropriate LOAD mouse models, but as noted recently, there is currently no single mouse model that recapitulates all the features of human AD [16]. This gap has led to an NIH NIA-funded program, Model Organism Development and Evaluation for Late-Onset Alzheimer’s Disease, or MODEL-AD (https://www.model-ad.org/, accessed on 30 December 2021), to develop mouse models of AD that better recapitulate the human disease. A focus of the consortium is to integrate into mouse models specific humanized risk variants, including various polygenic models overlaid on a background of aging. Some progress has been made in generating new models of LOAD that mimic aspects of AD [17], but there is currently no mouse model that reiterates the human disease, in part due to the multifactorial biological contributors.

This review summarizes the neuroimaging findings from two of the most studied AD mouse models, the 5xFAD and 3xTg-AD mouse models. As illustrated in Figure 1, both models have had and continue to have significant numbers of citations, with the 5xFAD becoming one of the most studied models. While MRI has not been used significantly, there have been emerging publications that use neuroimaging to track and understand disease progression, which are reviewed below. We start this review by briefly describing the different neuroimaging modalities that are utilized in human AD.

## 2. Human Imaging of Alzheimer’s Disease: Brief Overview

Within the clinical setting, neurological diagnosis of AD is performed by assessing memory impairment, thinking skills, and functional abilities. Additional insights into the progression of disease can be garnered by identifying behavioral changes that emerge in interviews with a patient’s friends and family. These observations are then combined with a range of risk factors (age, sex, family history, presence of APOE ε4, etc.) to enhance the diagnosis of AD. It should be noted that inaccuracy is frequent in the clinical diagnosis of AD, as shown by post-mortem histological studies [18,19]. More recently, genetic testing for the presence of a variety of known genetic risk factors has greatly assisted diagnosis, although there has not been a single gene or a unique set of genes to explain age-related cognition and ultimately AD [20]. Thus, when the neurological assessment for AD is not clear, neuroimaging techniques such as Magnetic Resonance Imaging (MRI), Computed Tomography (CT), or Positron Emission Tomography (PET) can be and are used to detect AD hallmarks. There are numerous extensive reviews available on human neuroimaging and AD [21,22]; herein, we briefly summarize the key findings.

### 2.1. Positron Emission Tomography (PET)

The original ^11^C-Pittsburgh Compound B (PiB) was developed to image Aβ deposition with high retention in brain regions known to be vulnerable in AD [23]. Considerable progress has been made since this original compound for imaging Aβ, with three clinically approved tracers now available. These new tracers are centered around ^18^F, which has a much longer half-life (hours) compared to ^11^C (minutes). The new tracers are Florbetapen (NeuraCeq^TM^), Flutemetamol (Vizamyl^TM^), and Florbetapir (Amyvid^TM^). While these PET reporters can identify patients with existing and advanced AD, they have also been useful in early-stage diagnoses [24], and may potentially differentiate dementia types [25]. These and other studies have revolutionized our understanding of Aβ deposition and progression, as well as its relationship to memory, cognition, metabolism, and structural alterations, as AD advances in human patients.

PET imaging of tau is relatively new and includes the FDA-approved tracer ^18^F-flortaucipir (Tauvid^TM^). This tracer binds to hyperphosphorylated tau proteins. Studies have shown that ^18^F-flortaucipir binding density is related to memory impairments and positively correlated with MRI-derived hippocampal volumes [26]. Recent studies with larger cohort sizes have concluded that tau PET imaging is useful as a biomarker for AD progression and staging [27,28].

Other PET imaging compounds have been and are being actively developed to target imaging of neuroinflammation, which is strongly associated with AD. Many of the clinical PET ligands for inflammation target activated microglia via the 18kDa translocator protein (TSPO). Lagarde and colleagues gave an excellent review on the current state of TSPO PET imaging [29], as well as others [30]. Other ligands targeting astrogliosis, such as ^11^C-Deuterium-L-Deprenyl (^11^C-DED), are being tested as predictive factors to detect AD prior to symptomology [31,32]. A recent in-depth review explores glial PET imaging [33]. Synaptic loss has also been evaluated in patients with mild cognitive impairment and AD using ligands for synaptic vesicle protein 2A (SV2A). ^11^C-UCB-J and ^18^F-UCB-J both highlighted hippocampal synaptic loss and correlated it with cognitive decline [34,35]. It is important to note the limitations to clinical and preclinical PET imaging that exist but are vigorously being addressed, including off-target binding, specificity as it relates to other neurodegenerative diseases, and the cost and availability of PET scanners. A broad review highlights the strengths and limitations of current PET ligands as they relate to human pathology [36]. Exciting new advances in analytic methods for PET data have led to molecular connectivity studies, which may bolster the ability to non-invasively monitor brain connectivity as AD progresses [37].

### 2.2. Magnetic Resonance Imaging (MRI)

MRI is the primary tool for brain imaging as it allows visualization of gray and white matter, cerebrospinal fluid, and ventricles. Broadly, in suspected AD cases, MRI is performed to detect cortical and/or hippocampal atrophy, ventricular enlargement and decreased brain volumes. CT can be used in cases where MRI is contraindicated for visualization of gross abnormalities, including hippocampal and cortical atrophy [38].

The earliest neuroimaging findings in AD from human subjects described a unique temporal pattern of abnormalities (topography). These structural abnormalities consistently encompassed the entorhinal cortex and hippocampal regions and were found to be predictive as patients progressed from MCI to AD [39]. Interestingly, as AD progresses, frontal cortices and thalamic regions also become more involved and increasingly impact the default mode network [40]. The hippocampal reductions are strongly mirrored by robust decrements in synaptic densities combined with abnormal synaptic morphologies [41]. Ongoing neuroimaging studies and analytics that evaluate radiological evidence and combine genetic testing have been advancing our knowledge. A recent study by Veldsman reported strong associations between sex (female small hippocampi) and APOE ε4 status that were linked to specific hippocampal sub-regions [42]. Thus, volumetric indices are strongly associated with AD and continue to be utilized as a surrogate marker.

MR imaging of brain networks, both structural (diffusion tensor; DTI) and functional (resting state fMRI; rsfMRI), has been utilized in the assessment of human AD. DTI lends itself to the assessment of regional microstructural integrity (or lack thereof) and is particularly well suited to the evaluation of white matter integrity [43]. In neurodegenerative diseases, such as AD, DTI has been especially useful in assessing gray and white matter brain pathology, as recently reviewed [44]. Luo and colleagues reported reduced DTI-derived fiber density in limbic tracts, corpus callosum and others in LOAD patients from two large separate databases [45]. These decrements in white matter were associated with amyloid PET imaging of the fornix. Emerging studies using DTI have identified a group of white matter tracts that are predictive for conversion to AD from MCI [43].

DTI metrics, such as mean (MD), axial (AxD), radial diffusivity (RD) and fractional anisotropy (FA), have, in general, reported decreased FA with increased AxD, RD and MD in white matter [46]. These authors noted that other acquisitions (i.e., multishell) could provide additional information for discriminating MCI from AD subjects. Indeed, neurite orientation dispersion and density imaging (NODDI) assessment confirmed decreased axonal density (neurite density index, NDI) and increased dispersion (orientation dispersion index, ODI) in MCI subjects [47]. Others have reported similar decrements in FA with increases in NODDI-derived isotropic water index (ISOVF), NDI and ODI that strongly correlated with Mini-Mental State Evaluation (MMSE) scores [48]. In gray matter, NDI/ODI were lower in AD patients, and in selected regions (i.e., temporal, parietal cortices) had the ability to discriminate between MCI and AD subjects [49]. A significant limitation of these and other studies relates to the small sample sizes and lack of post-mortem confirmation, but future conclusive studies will supplement these intriguing results.

Structural connectivity (DTI) allows the evaluation of connections between brain regions, particularly those in vulnerable brain regions. Loss of temporal lobe connectivity was associated with CSF phosphorylated tau and decrements in memory function, albeit in a small clinical cohort [50]. Graph-theoretical measures of connectivity, such as rich club (efficiency measure) connections, were disrupted in AD, and this disruption propagated from the peripheral regions (non-rich club) to cortical rich club regions (i.e., cortical and thalamic regions) [51]. Ensemble measures of DTI connectivity based on graph theory were robustly able to differentiate controls, MCI and AD subjects, consistent with some measures having a greater ability to discriminate disease states [52].

Functional connectivity from rsfMRI has also been utilized to identify subjects at risk for AD, as recently reviewed [53]. Similar to structural connectivity findings, rsfMRI connectivity was altered in the posterior cingulate cortex and precuneus area in MCI AD subjects [54] where decreased temporal lobe connectivity was related to AD [55]. Task-based fMRI studies have not been as prominent as rsfMRI, in part as task-oriented studies are designed to probe specific brain processes. For example, probing episodic memory, Nellessen and colleagues found in AD subjects a strong activation in the precuneus during encoding, but reduced activation in the right hippocampus during the retrieval phase, both of which were associated with lower cognitive scores [56]. Thus, task-based fMRI can be used to confirm the relationship between brain function and connectivity as it relates to AD.

There are several other MR-based imaging methods, including a wide array of analytic methods that can be used to assess AD. These include susceptibility-weighted imaging (SWI), magnetic resonance spectroscopy (MRS) and perfusion-based MRI to, name a few (see [22]). Intriguing advances in imaging transcriptomics are leading the way to identify genomic alterations that correlate to neuroimaging measures [57]. In summary, multimodal MRI for the study of MCI and AD provides diagnostic information on the progression of disease, whereby clinicians and investigators alike can select the appropriate imaging sequence(s) based on the question being probed.

Using the background from human AD imaging, we now review the preclinical literature in two popular mouse models of AD.

## 3. Mouse Models of AD

While the outcomes of clinical neuroimaging studies are limited by the difficulty of achieving a definitive AD diagnosis during a patient’s life, mouse models of AD have tremendously contributed to our biological understanding of AD pathophysiology. With the identification of the APP gene mutation as a contributor to AD, the initial models were mice with transgenic expressions of human APP. For example, the PDAPP model contains the V717F (Indiana) mutation, while the Tg2576 model contains a double mutation K670N/M671L (Swedish). A complete review of APP mouse models has been published [58].

Mouse models mimicking other human genomic alterations have been developed, including those containing PSEN1 mutation(s) that have been combined with APP, as in the hAPP/PSEN1 double transgenic model. For example, APPswe/PS1DE9 mice contain the hAPP Swedish mutation and the PSEN1 DE9 mutation [59]. The 5xFAD model, which was created to accelerate amyloid deposition in vivo, contains five combinatorial mutations (Swedish K670N/M671L, London V717I and Florida I716V in hAPP, M146L and L286V in PSEN1), leading to the rapid progression of disease [60].

Although these and other models have immensely helped the scientific community understand aspects of AD pathology, the need to evaluate the role of NFTs in AD has led to the development of additional mouse models containing tau mutations or insertions of human tau. The 3xTg-AD mouse model combines mutant hAPP (Swedish), PSEN1 (MM146V) and tau (P301L) transgenes, resulting in Aβ and tau pathologies [61]. The 3xTg-AD mouse exhibits more tau deposition compared to Aβ, as illustrated in Figure 2.

## 4. Neuroimaging of Mouse Models of AD

With advances in the clinical use of diagnostic neuroimaging, particularly MRI, it is not surprising that AD mouse models have also undergone preclinical MRI evaluations. A key advantage of preclinical MRI is the availability of contrast types that can assess brain structure and function. The most common MRI sequences used in preclinical AD imaging are summarized in Table 1. PET has also been used in mouse models to assess AD hallmarks such as Aβ and hyperphosphorylated tau accumulation, cerebral hypometabolism, glial activation and synaptic loss. We now summarize the key preclinical MR and PET findings in two of the most popular AD mouse models, specifically highlighting each imaging contrast.

### 4.1. 5xFAD

The 5xFAD mouse model of familial AD presents a fast progression of the disease and high expression of cerebral Aβ42 as early as 1.5 months of age, with amyloid deposition and gliosis evident at 2 months. Frank amyloid deposition at 9 months of age results in neurodegeneration accompanied by neuronal loss [60]. Presynaptic dystrophic neurites are also detected in areas surrounding amyloid plaques at 5–6 months [62]. There is a paucity of coherent neuroimaging publications that characterize and phenotype this AD mouse model across sex and over time, which has resulted in an incomplete neuroimaging characterization of this model. Table 2 summarizes the current state of neuroimaging studies for the 5xFAD mouse model.

#### 4.1.1. MRI: Volumetric

In the context of two sequential behavioral and cognitive studies, Girard and colleagues used MRI to assess volumes from several brain regions [66,68]. T2-weighted images (T2WI) were acquired from 2-, 4-, and 6-month-old 5xFAD and age-matched wild-type (WT) mice, and we found no significant differences in the manual derivation of volumes from forebrain, cerebral cortex, hippocampus, ventricles, striatum, olfactory bulbs or frontal cortex between 5xFAD and WT mice. The lack of male or female volumetric changes was in contrast to the appearance of behavioral deficits.

A PET/CT and MRI study with 13-month-old 5xFAD male mice revealed a 10% reduction in hippocampal volume compared to age-matched WT mice [69]. Like previous studies, no significant differences were observed at younger ages (2, 5-month-old).

#### 4.1.2. MRI: Morphologic

Spencer and colleagues used multimodal MRI to develop diagnostic markers of AD [65,73]. T1WI and T2WI, alongside histological staining, in 11-month-old 5xFAD mice revealed lower T1 relaxation times (s) in male/female 5xFAD mice. The reduced T1 values and increasing Aβ load were negatively correlated in the upper and lower cortex, white matter and hippocampal regions [65]. They then tested the hypothesis that T1 relaxation times could be used as a sensitive marker to detect Aβ load at early stages. However, T1 values were not significantly different between WT and 5xFAD at younger ages (2.5- and 5-month-old) prior to the onset of robust Aβ deposition [73]. At 11 months of age, there were no significant differences between WT and 5xFAD T2 values [65].

Susceptibility-weighted imaging (SWI) makes use of endogenous dephasing to detect iron content. Increased iron has been associated with amyloid plaques [90]. We have previously demonstrated that SWI provides excellent concordance between amyloid deposition and Prussian Blue staining for iron and SWI in the TgSwDI AD model [91]. A new study using in vivo manganese-enhanced MRI (MEMRI) as an activity-dependent contrast agent found that manganese could serve as a targeted contrast agent to visualize amyloid plaques in 8- to 9.5-month-old 5xFAD mice. Like our study, they also showed that even though manganese improves the signal-to-noise ratio, SWI alone was sufficient to detect plaques in high-resolution MR images [86].

Microvascular MRI is a technique that allows the mapping of the mean vessel diameter, microvascular density, and vessel size index [92]. It was developed in the late 1990s and uses fast gradient-echo spin-echo sequences combined with a paramagnetic contrast agent to report vascular features. This MR technique was used in 6-month-old 5xFAD male mice and revealed microvascular damage in the cortex of the transgenic mice but no damage in the hippocampus or thalamus compared to WT mice [87].

#### 4.1.3. MRI: Diffusion

We are currently unaware of any studies examining the 5xFAD model with diffusion MRI (dMRI) metrics, including single-shell diffusion tensor imaging (DTI) or multishell DTI approaches. This is a significant knowledge gap that we have begun to investigate. Figure 3 illustrates the DTI contrasts available in a 5xFAD mouse model acquired at 9.4T.

#### 4.1.4. MRI: Functional MRI and Connectivity Studies

Functional MRI studies utilize endogenous (blood oxygenation level-dependent; BOLD) or exogenous (e.g., manganese) contrasts to report neuronal activity changes in brain function. A functional study using MEMRI showed that T1WI signal intensity, as a proxy for neuronal activity, was associated with deficits in spatial cognition as assessed by Morris water maze (MWM) [70]. Neuronal activity was increased in 5xFAD males compared to age-matched WT mice. No differences were observed at 1 month of age, but neuronal activity was increased as early as 2 months in the hippocampus and entorhinal cortex. At 5 months, neuronal activity was also increased in the retrosplenial cortex and caudate putamen [70]. In analytic enhancements using a Paxinos and Franklin-derived atlas applied to the MEMRI data, the same group detected increased neuronal activity in the hippocampus and amygdala of 5xFAD male mice at 1-, 2-, and 5-months compared to age-matched WT [93].

Resting state functional MRI (rsfMRI) was performed in 5-month-old 5xFAD followed by connectomic analyses to characterize brain network organization in 5xFAD mice [75]. Connectomic analyses are used to model whole and regional brain networks using graph theoretical measures where interactions between nodes (brain regions) and edges (degree of connections) may define structural (DTI) or functional connections (rsfMRI or evoked fMRI) [94]. In male 5xFAD mice, the authors showed that structural networks exhibited higher path lengths and a lower small-worldness in 6-month-old 5xFAD mice compared to controls [75]. The in vivo functional networks were less conclusive in 5xFAD mice, and only demonstrated a higher path length in this pilot study. The authors concluded that the observed disconnectivity is similar to that reported in human studies.

#### 4.1.5. MRI: Spectroscopy

In vivo MR spectroscopy has been used to identify potential biomarkers of AD progression in the hippocampus of 5xFAD mice [63]. Proton spectroscopy (^1^H-MRS) was used longitudinally in WT and transgenic mice at 8, 9 and 10 months, and revealed decreased concentrations of N-acetyl aspartate (NAA) at 8, 9, and 10 months and decreased Gamma Aminobutyric Acid (GABA) at 9 and 10 months in the dorsal hippocampus of 5xFAD mice. Increased concentrations of myo-inositol (Myo) at 9 and 10 months were also observed. The decreases in NAA and GABA would be consistent with neuronal loss, which is reported late in life in the 5xFAD mouse [60]. Glutamate (Glu) and glucose concentrations were also reduced. In contrast to ^1^H-MRS, phosphorus (^31^P) spectroscopy at 8 and 16 months did not reveal significant differences between WT and transgenic mice [63]. Modeling of the ^31^P data suggested that there was no overt alteration in mitochondrial activity at the time points examined, but more recent work using alternate approaches would suggest a relationship between increased Aβ load and hypometabolism [83,89].

Similarly, Aytan and colleagues found decreased levels of NAA and GABA, but increased Myo and glutamine (Gln), in 8-month-old 5xFAD females when compared to age-matched controls [64]. A subsequent neuroprotection study in 3-month-old females found a large decrease in hippocampal taurine levels, as well as trending decreases in NAA and GABA, and significantly decreased Glu [71].

#### 4.1.6. PET: Fluorodeoxyglucose

There has been considerable interest in metabolic alterations in the 5xFAD mouse. ^18^F-FDG-PET in 13-month-old 5xFAD male mice revealed lower whole brain ^18^F-FDG uptake compared to WT controls [69]. Region-specific decreases were also significant in the amygdala, basal forebrain, basal ganglia, cerebellum, hippocampus, hypothalamus, neocortex and thalamus. Moreover, the WT and 5xFAD mice could be distinguished from each other using a regional brain glucose metabolism ratio that was predictive at ages as low as 2 months.

Similar results were reported in two additional PET studies of male 5xFAD mice [74,83]. ^18^F-FDG uptake was decreased in the whole brain of 5-, 7- and 12-month-old 5xFAD males compared to WT, and this was accompanied by significant regional decreases within the hippocampus, amygdala and thalamus [74,83]. Cerebellar glucose utilization was not significantly altered at 5 months [74] but was significantly reduced at 7 and 12 months in 5xFAD mice compared to WT [83]. Glucose metabolism was reduced in the cortex at 5, 7 and 12 months, but only significantly decreased at 5 and 12 months [74,83]. Another study using 9.5-month-old female mice reported a similar decrease in ^18^F-FDG uptake in the hippocampus and frontal lobe of 5xFAD mice compared to WT [77]. Interestingly, these reductions in brain metabolism would appear to be at odds with the unaltered mitochondrial activity as derived from spectroscopy (see above) [63].

However, conflicting results by Rojas and colleagues found an increased glucose metabolism in 11-month-old 5xFAD mice. It is unclear if sex may underlie these discrepant findings, as the gender was not reported. The authors suggested that these unexpected results could be due to increased inflammation and gliosis, which is prevalent at this age in 5xFAD mice, and thereby leads to an increased cerebral uptake of glucose [67]. Tataryn and colleagues in a recent paper found no significant differences in whole brain glucose uptake between WT and 5xFAD females at 7 or 12 months of age in contrast to the TgCRND8 mouse model [88]. This contrasts with the increased ^18^F-FDG retention in cortical regions of 12-month-old female 5xFAD mice compared to WT [89], possibly since mice in the Tataryn study were fasted prior to PET imaging. A clinical study demonstrated that increased plasma glucose levels, but not plasma insulin or insulin resistance levels, could explain the decreased ^18^F-FDG uptake [95]. As shown by Fueger et al., the biodistribution of ^18^F-FDG is influenced by fasting, body temperature, and the type of anesthesia used. These parameters should be standardized to allow for proper comparisons between future studies [96]. Additional studies are clearly needed to clarify these divergent metabolic studies.

#### 4.1.7. PET: Amyloid Imaging

PET amyloid imaging was performed in 10.5-month-old WT and 5xFAD mice using ^11^C-PiB, followed three weeks later by ^18^F-Florbetapir. The increased binding of both compounds (21% and 14.5%, respectively) in the brain of the transgenic mice was reported when compared to the controls (the authors did not report the sex of the mice) [67]. The rapid onset of amyloidosis in the 5xFAD model makes it attractive in assaying new PET tracers and new PET hardware. Frost and colleagues developed a small-animal PET/MRI hybrid system to assess amyloid deposition [82]. Using 14-month-old 5xFAD mice, they were able to detect an increased uptake of ^18^F-Florbetapir in the cortex, hippocampus, and thalamus relative to WT. The same tracer was used in 4- and 12-month-old male and female 5xFAD mice and could detect Aβ at 4 months with significant increases in Aβ accumulation by 12 months in both sex groups [89]. In an alternate PET tracer study, ^18^F-Florbetaben uptake was increased in the whole brain in both 7- and 12-month-old 5xFAD males [83]. Regional analyses found significantly increased regional uptake in the cortex, hippocampus, thalamus, forebrain, hypothalamus, amygdala and midbrain. A longitudinal ^18^F-Florbetaben PET/CT study in female 5xFAD and age-matched controls at 2, 5, 7, and 11 months also observed significantly increased uptake starting at 5 months of age in the cortex and hippocampus, but no differences in the brainstem [85].

These ^18^F-labeled amyloid tracers have lower cortical uptake and higher non-specific white matter uptake when compared to ^11^C-PiB. Thus, new PET ligands are being developed to find a more optimal amyloid PET tracer. ^18^F-FC119S is a new compound with a high selectivity and metabolic stability against in vivo defluorination. In humans, a study using ^18^F-FC119S showed a significant correlation with ^11^C-PiB uptake combined with a lower non-specific white matter uptake than ^11^C-PiB [97]. This new tracer was tested in 5.5-month-old 5xFAD males, and an increased ligand uptake was detected in the hippocampus, cortex, and thalamus, which was only significant in the hippocampus [78].

Cho and colleagues have designed a series of new ^64^Cu PET imaging agents with high specific affinity for amyloid deposits [84]. The longer half-life of ^64^Cu (12.7 h) makes these compounds particularly advantageous, especially for centers without on-site cyclotrons. When testing these ^64^Cu compounds, they found higher brain retention in 5xFAD mouse brains compared to WT, making these compounds potential candidates for diagnostic clinical PET for AD.

#### 4.1.8. PET: Other Tracers (TSPO, mGluR5, D2R, BChE…)

To the best of our knowledge, no tau-PET has so far been directly performed in the 5xFAD mouse model of AD. However, in a human study that found CSF-tau levels preceded positive tau PET, 5xFAD mice at 4, 6 and 12 months of age had elevated CSF-tau levels in relation to Aβ plaque loads [98]. Other PET tracers have been tested, including ^11^C-PBR28, to label 18kDA translocator protein (TSPO), which is known to label glial cells and has been utilized in a variety of disease states, including Huntington’s Disease [99]. In 6-month-old 5xFAD females, ^11^C-PBR28 exhibited significantly increased brain uptake compared to age-matched WT [72]. The mGluR5 tracer, ^18^F-FBEP, showed significantly decreased binding in 5xFAD mice compared to age- and sex-matched controls [76]. This was only observed in 10-month-old males, but while also decreased, it was not significant in 10-month-old females [80]. While sex was not specified, 9-month-old 5xFAD mice had a significantly decreased uptake of ^18^F-FBEP in the cortex, striatum, hippocampus, and thalamus, with no differences observed in the cerebellum [81].

While the predominant focus in AD PET imaging has been tau or amyloid, dopamine PET imaging was recently tested in 10-month-old 5xFAD females using ^18^F-fallypride (Dopamine D2 receptor tracer). ^18^F-fallypride exhibited significantly decreased binding in the brain of 5xFAD mice compared to WT, and primarily within the striatum [80]. Butyrylcholinesterase (BChE) PET imaging was also tested in 5xFAD mice by Rejc and colleagues [85]. BChE (as well as AChE) levels were increased in brain tissues where amyloid and tau levels were elevated, where BChE becomes the primary degradation enzyme for acetylcholine instead of AChE [100,101]. The novel ^11^C-labeled BChE’s inhibitor uptake was increased in the brain of female 5xFAD mice compared to WT, at 4, 6, and 8 months, but was equivalent between groups at 10 and 12 months [85]. The authors suggest that due to its increased binding at a young age in the 5xFAD mice, ^11^C-BChE could be an early prognostic biomarker for the development of AD.

### 4.2. 3xTg-AD

The triple transgenic model (3xTg-AD) was the first transgenic mouse model of AD that developed both amyloid plaques (Aβ) and NFTs in brain regions associated with AD pathology (Figure 2). Extracellular Aβ deposits and synaptic dysfunction are apparent at 6 months, with the appearance of NFTs at 12 months [61]. As noted in Figure 1, the 3xTg-AD mouse model is still popular, although with the emergence of the 5xFAD and other new models, it has not been used as extensively. In Table 3 we summarize the key neuroimaging studies using the 3xTg-AD model and their findings.

#### 4.2.1. MRI: Volumetric

Hohsfield and colleagues, using ex vivo fast low-angle shot (FLASH) T2WI, found increased lateral ventricular volumes in 14-month-old 3xTg-AD males compared to WT males [107]. Similar results were also reported using the radial diffusivity (RD) maps from DTI to acquire lateral ventricle volumes, wherein 2-month-old 3xTg-AD females exhibited lateral ventricular enlargements (4-fold) compared to controls [118]. Using in vivo T2-weighted rapid acquisition with relaxation enhancement (T2RARE) imaging, Algarzae and colleagues found a 21% increase in ventricular volume in 18-month-old 3xTg-AD mice compared to age-matched controls [102]. Cortical atrophy with observable brain lesions has been reported in 18-month-old 3xTg-AD mice [102].

More recently, Guëll-Bosch et al., also using T2RARE imaging, acquired volume measurements from the cortex, hippocampus, cerebellum, olfactory bulb, and the whole brain in 5-, 7-, 9-, and 12-month-old female 3xTg-AD mice [120]. The 3xTg-AD females had significant reductions in all regions except for the cortex at the 9-month time-point. The olfactory bulb volumes were reduced at the 7-, 9-, and 12-month-old time-points. Chiquita and colleagues also reported reduced hippocampal volumes in 3xTg-AD mice using T2RARE imaging [117]. The hippocampal volume reduction was increasingly accentuated over time, whereby voxel-based morphometry showed reduced gray matter volumes bilaterally in the hippocampus across all time-points. Interestingly, as AD progressed, the gray matter reductions were observed to propagate to more anterior regions, including CA1, CA2, CA3, and the dentate gyrus [117]. In vivo 3D-FLASH imaging reported increases in “local” (regional) brain volumes in 3xTg-AD mice from 2 to 4 months of age, followed by subsequent decreased volumes at 4 to 6 months [119]. Significant genotype interactions at 6 months of age were observed in the bilateral amygdala, left hippocampus, left ventricle, bilateral insular region, left septum, right nucleus accumbens, right fimbria, globus pallidus, bilateral anterior commissure, right inferior olivary complex, and right stria terminalis. In a study by the same group, 1.5-, 2-, 3-, 4-, 5-, and 6-month-old male and female 3xTg-AD mice exhibited significant volumetric decreases, when compared to controls, bilaterally in the dentate gyrus, piriform cortex, paravermal regions, inferior and middle peduncles, flocculonodular regions, fimbria, internal capsule, and corpus callosum, as well as in the left hippocampus and pontine nucleus. Volumetric increases were noted in the entorhinal cortex, bilateral amygdala, thalamus, striatum, nucleus accumbens, superior cerebellar peduncles, anterior commissure, and longitudinal fasciculus. The volumetric trajectories of the bilateral hippocampus, entorhinal cortex, frontal regions and fimbria in 3xTg-AD mice took on an inverted parabola [115]. While the authors acknowledge the potential sex effects in their study, they did not explicitly test for this.

It is important to note that while most studies found brain-wide volumetric reductions, several studies reported no significant volume differences between 3xTg-AD and WT mice. In 6-month-old female mice, no statistically significant differences in total brain volume between 3xTg-AD and WT mice were observed [114]. Regional volumes of white matter structures were acquired in 11-, 13-, 15-, and 17-month-old 3xTg-AD and age-matched WT mice, and from both in vivo and ex vivo MRI, but no statistically significant differences were found in the appearance and volume of the corpus callosum, external capsule, and fornix at different ages between the 3xTg-AD and WT mice [104]. Similarly, Wu and colleagues reported no significant differences in cortical or hippocampal volumes acquired from in vivo T2-weighted images in 22-month-old mice [108].

Differences between studies that documented either decrements or no changes in whole brain or regional volumes could be due to a variety of factors (see Table 3). These include magnet strength, imaging sequence, type of post-processing, and whether manual regions of interest or atlases were used to extract regional data. For example, those studies that found no significant changes in volumes reported manual analyses or data reported as percent of controls [104,108,114]. Finally, it is important to note the background and lineage of the 3xTg-AD mice (i.e., number of generations) used in the reported studies, as disease models are vulnerable to genetic drift [124].

#### 4.2.2. MRI: Blood–Brain Barrier Permeability

Blood–brain barrier (BBB) pathology is well known to be involved in the pathogenesis of AD, where BBB breakdown promotes disease progression. MRI, in conjunction with exogenous contrast agents (i.e., gadolinium; Gd), can monitor the status of the BBB in vivo. Several MRI studies report conflicting BBB breakdown results in the 3xTg-AD mouse model. Chiquita and colleagues assessed BBB permeability using 2D dynamic contrast-enhanced FLASH with injection of a Gd contrast agent (Gadovist^TM^) [117]. In 16-month-old 3xTg-AD mice there was an increased BBB permeability index, and a decreased perfusion peak amplitude, indicating decreased vascular volume, compared to WT mice. In contrast, Ishihara did not observe any changes to BBB permeability in 6-month-old 3xTg-AD mice using T1-weighted spin-echo multi-slice imaging enhanced by a Gd contrast agent (Gd-HP-DO3A; ProHance^TM^) [103]. These studies would suggest that at early ages, the 3xTg-AD mouse does not exhibit BBB alterations, but with advancing age and AD pathology, the BBB is disrupted.

#### 4.2.3. MRI: Morphological

A number of studies have shown that one of the hallmarks of AD, the presence of amyloid plaques, can be observed using MRI. Dudeffant and colleagues used in vivo and ex vivo 3D gradient echo imaging combined with a Gd contrast agent to detect amyloid plaques [113]. Aβ-stained histological sections were co-registered with MR images to ascertain the identity of potential amyloid deposits. In ~11–18-month-old 3xTg-AD males, intracellular amyloid deposits detected using histology were not observed in Gd-enhanced MRI. In 3xTg-AD males 18–36 months of age, amyloid plaques were MRI-detectable, but only in the subiculum. The authors concluded that Gd-enhanced MRI exhibited poor detection of amyloid plaques, as only the largest plaques were detectable [113]. In another study, T2 relaxation measurements were higher in the hippocampus and cortices of 5- and 7-month-old 3xTg-AD compared to WT mice [120]. Increased T2 values may report neuroinflammation and neurodegeneration. In both groups of mice, T2 values were similar at the 9-month time-point and tended to decrease over time, where reductions in T2 values were associated with the increased presence of amyloid plaques [120].

At the present time, SWI has not been implemented in 3xTg-AD mouse models. Clearly, additional research is warranted to determine the utility of specialized MR sequences for assessing amyloid plaques or putative signals related to amyloid deposition.

#### 4.2.4. MRI: Diffusion

DTI has started to be used to evaluate tissue microstructure in several brain regions of 3xTg-AD mice alongside WT controls. No white matter DTI changes were found in any metrics from 3xTg-AD mice with ages spanning 11 to 17 months [104]. The authors also reported no changes in myelin staining or white matter volumes. An additional study using 12- and 14-month-old mice found DTI metric changes (decreased FA and AxD, increased RD) in the hippocampus, but not in the cortex [112]. The authors used histology and suggested that the changes they observed might be due to the presence of Aβ plaques and tangles in the hippocampus, while the cortex only presented with a few tau-positive neurons. In younger (2-month-old) 3xTg-AD mice, Manno and colleagues found increased cortical FA with reduced RD, but no changes in MD or AxD [118].

DTI was used to evaluate tissue microstructure in several brain regions in 3xTg-AD mice [121]. DTI in 3xTg-AD mice at 2 months of age found reduced AxD and FA, with increased radial kurtosis in the basal forebrain, which exhibited progressive temporal declines in AxD and FA at 8 and 15 months of age. In the hippocampus, elevated DTI metrics were observed (MD, AxD, RD, FA) at 2 and 8 months with no changes at 15 months in 3xTg-AD compared to controls. No diffusion kurtosis imaging (DKI) measures changed in the hippocampus. Similar DTI increases at 2 and 8 months of age in the 3xTg-AD were reported in the fimbria but not the fornix. The authors noted consistent 3xTg-AD vs. WT control differences throughout their lifespan [121]. However, the interpretation of these diffusion metrics changes is difficult/complex, and would need to be associated with histological staining to assess the changes in cellularity in the different brain structures.

#### 4.2.5. MRI: Functional MRI

Only a single 3xTg-AD study utilized functional connectivity via rsfMRI [118]. The study reported reduced hippocampal interhemispheric, but no change in caudate putamen functional connectivity in 2-month-old 3xTg-AD mice compared to WT mice. Even at early ages, hippocampal interhemispheric connectivity was reduced prior to the robust onset of pathology [118].

#### 4.2.6. MRI: Spectroscopy

Magnetic resonance spectroscopy (MRS) has been extensively used to detect biochemical changes in the 3xTg-AD mouse model. In vivo ^1^H-MRS found a significant reduction in taurine in 3xTg-AD mice compared to age-matched controls in the 4-, 8-, and 16-month-old groups [117]. The relative taurine levels are postulated to be related to the degree of neuroprotection in AD [81]. In addition, metabolic profiles (in vivo ^1^H-MRS) of the hippocampus and cortex in 6-month-old 3xTg-AD mice found only increased alanine levels compared to controls [114]. In 5-month-old 3xTg-AD mice, increased phenylalanine was observed, but taurine was decreased in the hippocampus compared to controls. At 7-months of age, the AD mice showed an increase in Myo that was still present at 12 months of age [120]. The authors also noted decreased NAA in the cortex, as was reported in the 5xFAD model [64,71,120]. Using ex vivo ^13^C-MRS after an infusion of [1-^13^C]-glucose in 7-month-old 3xTg-AD and WT mice, there was a 50% increased influx of ^13^C in 3xTg-AD mice compared to controls [106]. The authors infer that these increases were a measure of hypermetabolism in 3xTg-AD mice. Specifically, increases were found in [4-^13^C]Glu, [3-^13^C]Glu, [1-^13^C]Glu, [4-^13^C]Gln, [3-^13^C]Gln, [2-^13^C]Gln, [1-^13^C]Gln, [2-^13^C]Asp, [4-^13^C]GABA, and [1-^13^C]GABA [106]. In contrast to the findings summarized above, Carreras and colleagues did not find any significant differences in any spectroscopic metabolites in 7-month-old 3xTg-AD mice using ex vivo ^1^H-MRS in hippocampal tissue punches, but these findings were not compared to WT mice [125].

#### 4.2.7. PET: Fluorodeoxyglucose

Regional analysis of glucose uptake was measured using ^18^F-FDG-PET in 7-month-old 3xTg-AD male mice, with no significant differences in ^18^F-FDG uptake compared to WT controls in the bilateral hippocampus, or in the motor and somatosensory cortex [106]. However, the authors did observe a decrease in ^18^F-FDG uptake in the whole brain compared to WT. While sex was not specified, an earlier study by the same group similarly found a decline in ^18^F-FDG uptake in the whole brain of 7- and 13-month-old 3xTg-AD mice compared to WT, with a greater reduction noted in the 13-month-old mice [105]. The authors suggested a deregulation of glucose uptake in 3xTg-AD mice that becomes more prominent with age. Another study also reported a significant reduction in cerebral glucose uptake in 16-month-old 3xTg-AD male and female mice compared to WT, with more prominent differences for females [122].

Adlimoghaddam and colleagues observed reduced ^18^F-FDG uptake in the bilateral piriform area and insular cortex of 11-month-old 3xTg-AD male mice compared to WT, but no differences in the whole brain [116]. Two additional PET studies using 6-month-old 3xTg-AD mice reported significant decreases in ^18^F-FDG uptake in the midbrain, diencephalon, hippocampus, and prefrontal cortex compared to WT [109,110]. A parallel study in 6-month-old 3xTg-AD male mice also reported a significant decrease in glucose uptake in the hippocampus of 3xTg-AD male mice relative to WT [111].

#### 4.2.8. PET: Amyloid Imaging

Chen and colleagues measured ^11^C-PiB uptake in 5-, 8- and 11-month-old 3xTg-AD mice and age-matched controls, with a significant increase in uptake in the whole brain of transgenic mice at 5 and 8 months compared to WT [123]. However, a conflicting study found no significant differences between cerebral ^11^C-PiB uptake in 4-, 8-, 12-, and 16-month-old transgenic and WT male mice. The authors note that PET imaging is a less sensitive measure of amyloid load, given that Western blot data showed increased levels of Aβ monomers for the 3xTg-AD mice relative to WT at all time-points [117].

#### 4.2.9. PET: Other Tracers (TSPO, HDAC)

Microglia activation in 3xTg-AD mice has been tested by PET imaging using ^11^C-PK11195 (TSPO tracer), and no significant differences were found between groups when examined across their lifespan (4–16 months) [117]. The authors proposed that microglia activation may not be a prominent feature in this transgenic model, but this is at odds with histopathological studies, where increased microglia were reported in the cortex (but not hippocampus) [126].

Class IIa histone deacetylase (HDAC) PET imaging was also tested by Chen, using the HDAC4 and HDAC5 tracer ^18^F-TFAHA. The results show that the uptake in the whole brain of 8- and 11-month-old transgenic mice was 1.15- and 1.63-fold higher than in the WT, respectively [123]. The 11-month-old transgenic mice showed an almost 2-fold (significant) increase in ^18^F-TFAHA uptake in the striatum, cortex, hippocampus, basal forebrain, thalamus, hypothalamus, amygdala, and cerebellum relative to WT. Supportive cell culture uptake studies demonstrated that Class IIa HDAC uptake corresponded to Aβ levels [123].

## 5. Magnetic Resonance Findings in Other AD Mouse Models

### 5.1. Volumetric MRI

The early neuroimaging of AD mouse models focused predominately on volumetric alterations. One of the earliest studies in the PDAPP mouse (and its variants) found decreased hippocampal and dentate gyrus volumes [127,128]. Many AD mouse models have reported decreasing brain volumes, such as the APP T7141 [129]. The APP/PS1 model has morphometric abnormalities localized to the hippocampus, cortex, olfactory bulbs, stria terminalis, brain stem, cerebellum, and ventricles [130], while cortical thickness was also reduced in APP mice, which is exacerbated with age and Aβ deposition [131]. Interestingly, young (1-month-old) APP mice had increased cortical thickness, but the rate of cortical thinning was accelerated over the next 15 months of age compared to controls [132]. An unbiased approach, voxel-based morphometry (VBM), found age-related atrophy in the hippocampus, motor cortex, striatum, amygdala, septal area, bed nucleus of the stria terminalis, and nucleus accumbens, in APP/PS1 transgenic mice [133]. Badea and colleagues found that deficits in cortical volume precede amyloid formation in the APPswe/ind mouse model, similar to the presymptomatic atrophy seen in patients with familial AD [134].

Unlike humans, WT mice have increasing brain volumes between 6 and 14 months of age, but mice APP/PS1 (expressing human APP(695(K595N, M596L)) x PS1(M146V)) had global reductions in gray matter-rich areas that progressively declined with age (i.e., hippocampus) similar to humans [135]. A follow-up study in TASTPM mice compared to triple transgenic mouse model (APP/PS2/Tau) found that hippocampal atrophy was age-dependent, whereas in TASTPM mice, it was already detectable at the first investigated time point (<6 months of age) [136]. Importantly, both transgenic mouse models displayed an age-related entorhinal cortex thinning and robust striatal atrophy (associated with a significant loss of synaptophysin) [136]. The rTg4510 transgenic mouse, an increasingly popular AD model, reported severe hippocampal and neocortical atrophy with concomitant increased ventricles [137], where females had larger volume decrements than males and both sexes were reduced significantly compared to WT [138]. Other AD mouse models, such as ApoE4 (24 months), Tau4RDeltaK and Tg4–42 mice, also exhibited significant hippocampal and whole brain volume decrements [139,140,141].

Despite the numerous studies reporting volumetric regional and whole brain decreases in a variety of AD mouse models, several studies have either reported increased growth rates (such as in the TgCRND8 compared to controls) [142] or no volumetric changes in APP mice [143]. APP+SOD vs. APP transgenic mice showed robust brain atrophy [143]. Interestingly, the TgCRND8 mouse was reported to have increased growth rates in brain regions that contained the greatest density of amyloid plaques and reactive astrocytes, a finding that will require additional study [142].

In brief, the overwhelming evidence points to regional and whole brain volumetric changes in mouse brains of AD models using MR assessments, and the rate of change is driven largely by the underlying neuropathology in many of these models.

### 5.2. Magnetic Resonance Spectroscopy (MRS)

Early work using MRS to assess APP/PS1 mice found decreased NAA/Glu ratios [144], decreased NAA with increased Myo/creatine (Cr) ratios [133,145], reduced NAA and Glu levels in 5- and 8-month-old mice [146], and NAA that was decreased in the hippocampal as well as the temporal cortex, but Myo that was increased throughout the entire brain [147]. MRS was able to discriminate between APP/PS1 transgenic mice and WT mice as early as 2.5 months of age based on NAA, Glu, and macrolipids levels [148]. GluCEST imaging in APP mice documented reductions in Glu concentrations, with a ~30% reduction in all brain areas, which was confirmed with ^1^H-MRS [149,150]. Hippocampal NAA reductions in APP(Swe)/PS1(dE9) transgenic mice were related to the loss of CA3 neurons [151], and NAA/Cr reductions were concomitant with amyloid deposition [152]. Combining APP/Tg2576 genotypes resulted in mice with decreased whole brain metabolites of NAA, Glu and glutathione, which was ascribed to neuronal loss; increased taurine (glial) was also reported [153]. Thy-1-APPSL mice had reduced NAA/Cr ratios with decreased phosphorylcholine (PC) [154]. Glu and Gln were elevated, while GABA levels were increased, leading the authors to suggest that the Thy1-APP(SL) mice had mitochondrial dysfunction [154].

Various genetic combinations of AbetaPPswe with other transgenic mice resulted in similar metabolite reductions as reported in the APP mouse. AbetaPPswe/Tg2576 mice at 1 month of age had decreased concentrations of Glu and NAA in the hippocampus and rhinal cortex compared to WT mice [155]. With increasing age, other brain regions exhibited similar reductions in NAA and related metabolites, and by 11 months, increases in taurine and creatine were observed that have been ascribed neuroprotective functions. These metabolic perturbations were apparent prior to the appearance of behavioral disorders [155]. Van Duijn reported that AbetaPPswe and PSEN1dE9 transgenic mice had a similar trajectory of reduced metabolites, but the decline was more apparent in female mice, particularly for taurine [156]. In AbetaPPswe-PS1dE9 mice, hippocampal NAA levels were only reduced at 12, but not at 8 months of age [157].

Several other models also reported similar reductions in metabolites. The TASTPM mouse showed significant differences in Cr, Glu, NAA, choline-containing compounds, and Myo compared to WT mice [158]. An interesting study in female Tg2576 mice found elevated hippocampal GABA levels, which were associated with increased reactive astrocytes and amyloid deposition [159]. The authors suggest that regional GABA levels may contribute to the sex disparities observed in AD; but this awaits further study. In a recently developed tau mouse model of AD, Tg4-42, reduced 4-aminobutyrate, Gln and lactate in the cortex were reported, with reductions in 4-aminobutyrate in the caudate putamen [139]. These reductions mirrored increased levels of metabolizing enzymes and increasing Aβ42 levels with increased neuroinflammation and neuronal loss [139]. The Tg4-42 mouse replicates the progressive neuronal loss observed in human AD subjects.

MRS assessments, both global and regional, appear to identify losses of key brain metabolites as AD progressively develops in these mouse models. In some studies, it had the ability to discriminate early in life between the transgenic model and the WT before an overt pathology was evident. Thus, MRS in combination with other MRI techniques could potentially accurately identify at-risk human subjects.

### 5.3. Vascular MR Imaging

There is some surprising literature on MR imaging of the vasculature in AD mouse models, signifying the importance of blood vessels in AD pathology. Early MR angiography of APP23 mice found perfusion deficits that reflected progressively decreased vascular density with age, as assessed by synchrotron CT vascular corrosion casts [160,161]. This group extended these studies to show decreased cerebrovascular responses to evoked functional MRI (fMRI), confirming compromised vascular responses [162]. In the thalamus, the APP23 mouse exhibits cerebral amyloid angiopathy that leads to altered CBF; these perturbations were also observed in APP51/16 mice [163]. The intravenous administration of iron oxide particles showed significant uptake into macrophages within the cortex and thalamus in aged APP23 mice, which coincided with the loss of signal (increased nanoparticles deposition) predominately in vessels, suggesting overt microvascular pathology, a finding that was also reported in APP24 and APP51 mice [164]. Using MR-derived measures, microvascular density, mean vessel diameter and size index were examined in 3-, 6-, 9-, 14-, and 20-month-old APP23 and WT mice [165]. By 20 months, APP23 mice exhibited decreased microvascular density with increased vessel diameters compared to WT mice. The hippocampus of these APP23 mice exhibited abnormal microvascular angiogenesis early in life prior to large decreases, suggesting a potential compensatory response [165].

APP/PS1 transgenic mice exhibited significantly reduced cerebral perfusion, even at a young age [132]. A more comprehensive perfusion study confirmed significantly decreased perfusion in the left hippocampus, left thalamus, and right cortex of APP/PS1 transgenic mice [166]. Other variants had either reduced cerebral perfusion (APPxPS1-Ki) [167] or flow voids within the middle cerebral artery in older mice (APP/PS1 APP(SweLon)/PS1(M146L)) [168]. In contrast, no autoregulatory changes in APP were reported in this single study [169]. MRI measures of vascular morphology in the APP.V717IxTau.P301L: biAT mice at 3, 6, and 12 months of age showed increasing vessel radius or length with age regardless of genotype, but the biAT mice had significantly lower internal carotid artery length, although the significance is not immediately clear [170]. Reductions in perfusion and blood volume were observed in the occipital cortex of B6.PS2APP mice [171]. Several studies using the AbetaPP/PS1 mice found that increased systolic blood pressure was related to decreased regional CBF [172], but a related variant, the AbetaPPSWE/PS1DeltaE9 mouse, exhibited increased thalamic CBF at young ages (2–4 months) that was correlated with increased vessel area [173]. Similarly, in 13-month-old APP(swe)/PS1(dE9) mice, there was a decreased level of rCBV that was correlated with a capillary density consistent with altered cerebrovascular reactivity [174].

Tg2576 mice exhibited no apparent hemodynamic changes, although ^18^F-FDG-PET hypermetabolism was found in 7-month-old mice [175]. This was contrasted by MRA imaging by 17.6T in Tg2576 mice, which found severe blood flow defects in the middle and anterior cerebral arteries that may be related to vessel Aβ levels [176]. Cerebrovascular reactivity was increased in cortical regions with tau pathology in ~9-month-old rTg4510 mice [177], but showed global gray matter hypoperfusion [178]. In the hippocampus of TgCRND8 mice, there was decreased CBF that was related to decreased microvasculature length; similar findings were reported in APP/PS1 and P301S Tau-transgenic mice [179]. The decreased CBF in 12-month-old TgCRND8 mice was confirmed and reflected by reduced glucose metabolism without changes in cerebral blood volume [88]. Finally, the Tau4RDeltaK mouse model showed no change in CBF, which the authors ascribed to normal vascular function despite reduced global oxygen extraction fraction and cerebral metabolic rate [140].

The overwhelming finding in these transgenic models of AD was that there are dramatic reductions in the cerebral perfusion. These age- and pathology-related decreases have been suggested to precede the behavioral manifestations and, in some instances, even come prior to tissue-level pathology. Given the significance of human vascular-related AD and dementia, more research in emerging mouse models of AD that better reflect the human condition is warranted.

### 5.4. Diffusion MRI (dMRI)

Diffusion MRI encompasses a range of diffusion imaging methods, from simple diffusion-weighted imaging (DWI) to directionality-driven diffusion tensor imaging (DTI), and more recently to multishell variants, to better model the underlying tissue architecture. APP23 mice showed reduced DWI in regions with high Aβ deposition [180]. In 17–25-month-old APP23 mice, there were sex-related differences, wherein females showed larger decreases in extracellular space and apparent diffusion coefficient (ADC) than in males, but this was not found in younger mice and was attributed to a larger Aβ plaque load in females [181].

A considerable number of DTI assessments have been performed in APP/PS1 mice and their variants. No changes in FA, MD, AxD and RD in 8-month-old APP/PS1 mice were observed [166]. However, earlier studies using voxel- and atlas-based morphometry found that in 14-month-old APP/PS1’s gray (neocortex, hippocampus, caudate putamen, thalamus, hypothalamus, claustrum, amygdala, and piriform cortex) and white matter areas (corpus callosum, external capsule, cingulum, septum, internal capsule, fimbria, and optic tract), there was increased FA and AxD [182]. At least in the APP/PS1 mice, this would suggest advancing brain alterations. A similar study found that cingulate cortex and striatum in APP/PS1 mice had significant FA and AxD increases, while the thalamus only had increased FA [183]. The authors surveyed other regions, finding that MD, AxD, and RD increased in the bilateral neocortex, with only the left hippocampus exhibiting increased FA compared to WT. White matter regions showed selected regional increases in FA or AxD (forceps minor of the corpus callosum, anterior part of the anterior commissure, internal capsule) [183]. As would be expected with increasing age and neuropathology, these mice demonstrated altered DTI metrics in brain-wide (gray/white matter) regions. Interestingly, the APP/PS1 (APPKM670/671NL/presenilin 1 L166P) variant at 16 months of age displayed no change in DTI metrics, but DKI metrics were increased in cortex and thalamus, consistent with the known improved sensitivity in DK imaging [184]. In 12-month-old APP(swe)/PS1(dE9), AxD and RD were reduced in most white matter tracts (corpus callosum, fimbria of the hippocampus) and MD, RD and AxD were increased in the hippocampal region [185].

The 12- and 18-month-old Tg2576 mice showed RD and AxD reductions in their gray and white matter when appreciable amyloid plaque accumulation was confirmed, while the corpus callosum showed decreased AxD but elevated RD, which the authors related to Aβ load leading to demyelination [186]. Another study found reductions in RD in 12–17-month-old Tg2576 mice, but surprisingly these were not significantly different from WT mice [187]. The 8–9-month-old rTg4510 exhibited tau-related atrophy in gray matter, but only showed increased RD in white matter structures [178]. The rTg4510 mice at 2.5, 4.5, and 8 months underwent DTI, but only in 8-month-old rTg4510 mice was there decreased FA in white matter regions (corpus callosum, anterior commissure, fimbria, and internal capsule), which was accompanied by higher RD when compared to WT mice [188]. NODDI processing of DTI acquisitions in rTg4510 at 8.5 months of age found that the neurite density index (NDI, the volume fraction that comprises axons or dendrites) reflected the histological measurements in gray matter, but they also found decreased FA and orientation dispersion index (ODI) in white matter [189].

TgCRND8 diffusion imaging in 12–16-month-old mice noted global reductions in ADC in transgenic mice, but these were not significant compared to WT mice [190]. A more comprehensive study in TgCRND8 mice using DTI reported white matter and hippocampal RD and AxD deficits [191]. Using NODDI analytics, these authors found increased ODI and intracellular volume fraction (ICVF) in the white matter and hippocampus, consistent with more complex axonal and dendritic processes. Male and female hAPOE4 mice exhibited lower FA than their hAPOE3 littermates, suggesting a lower level of white matter integrity in hAPOE4 mice [192]. Finally, in the CVN-AD mouse model, it was found that of all the DTI metrics, FA was most sensitive, then RD, followed by AxD, in the detection of hippocampal pathological changes (atrophy). DTI white matter reductions were largest in the fornix (23%) [193].

While the DTI findings from the myriad of mouse models of AD were not overly consistent, there were notable changes in AxD, RD and FA within the white matter of many of these mice, particularly at older ages. One feature that does emerge is that DTI can identify altered diffusion in gray matter, which in some studies proceeds neuropathology. NODDI metrics also exhibit improved sensitivity over standard DTI, but more studies are needed to confirm its diagnostic capabilities. Unlike some of the other imaging methods, DTI data can be processed using a number of different software packages, and with a wide range of settings that can ultimately influence the outcomes. At the present time there is no formal consensus on what the best practice for DTI/NODDI acquisitions and analyses should be.

### 5.5. Relaxometry Imaging

Tissue relaxometry can provide insights into the underlying tissue structures, for example, increased water content can be assessed using T2WI. Several studies have investigated tissue relaxation, although this is often not the primary focus of the MRI studies. In B6.PS2APP mice (10–17 months), significant decreases in T1 and T2 relaxation were recorded in frontal cortices and in subiculum/parasubiculum regions, in which histologically confirmed presence of plaques [171]. Similarly, decreased T2 values were found in the hippocampus, cingulate, and retrosplenial cortex of 16–23-month-old PS/APP mice, but not in the corpus callosum [194]. A study of three transgenic genotypes (APP, PS/APP, and PS; >12 months of age) reported that all three models exhibited decreased T2 relaxation values in the cortex and hippocampus, and that amyloid deposition was correlated with the largest T2 reductions [195]. In APP/PS1, the mean T2 values decreased with age in the hippocampus, cortex, corpus callosum, and thalamus, but significantly increased T2 values were found in the hippocampus, corpus callosum, and thalamus in younger APP/PS1 compared to WT [196]. The reverse was true for aged APP/PS1 mice, with significantly decreased T2 values in the hippocampus and cortex relative to WT.

Increased T2 signal intensity was reported in the cerebral cortex of 12-month-old AbetaPP/PS1 mice [197], but in a different study in 5-month-old mice, a decreased T2 relaxation in the neocortex, caudate, thalamus, hippocampus and cerebellum was observed without histopathological changes in gray matter density [198]. This highlights the need for quantitative multi-echo relaxation MR imaging for tissue-specific properties in AD mice, as signal intensity changes are vulnerable to many factors.

Two studies were performed across the life span of Tg2576 mice [199,200]. White matter (corpus callosum) showed progressive increases in T2 values as mice aged from 10 to 18 months, which were accompanied by hippocampal and cortical T2 decreases [199]. Histology confirmed significant demyelination, gliosis and amyloid plaque deposition in the corpus callosum. Roy and colleagues focused on the suprachiasmatic nucleus (SCN), a region important for sleep and circadian rhythms known to be impacted during aging [200]. In vivo T2 relaxation decreased with age (10–18 months) in the SCN, which was greater in females compared to males and aged-matched controls. Within the SCN, elevated reactive astrocytes and an increased astrocyte/neuron ratio were hypothesized to underlie the T2 reductions [200]. Lastly, the TASTPM mouse exhibited reduced T1/T2 relaxation values, from which the authors concluded that the transgenic mice had altered water content [158].

In summary, the use of T1 and T2 relaxometry imaging could be useful for the rapid assessment of underlying pathology. At present, it is unlikely that T1/T2 relaxation would be overtly predictive for AD onset but appears to reflect developing or existing pathology.

### 5.6. fMRI

fMRI can be used to map connectivity or response to stimuli throughout the brain. Grandjean showed that PSAPP mice had mild functional responses within the supplementary and barrel field cortices, which was associated with amyloid deposition, but these functional changes were not found in E22ΔAβ transgenic mice [201]. The ArcAβ mouse had strong functional connectivity changes between brain regions that were more apparent than the PSAPP mice. ArcAβ mice had greater intracellular Aβ aggregates with increased parenchymal and vascular amyloid deposits, leading the authors to conclude that strain-specific functional and structural changes are not necessarily directly related to amyloid deposition [202]. In another study, APPNL mice were compared with APPNL−G−F (Swedish, Iberian and Arctic APP mutations) at 3, 7 and 10 months of age to assess rsfMRI connectivity between hippocampal and prefrontal regions. The only significant change was in 3-month-old APP(NL-G-F) mice, which exhibited increased prefrontal–hippocampal network synchronicity, and which occurred prior to robust amyloid deposition [203]. Additionally, there was hypersynchronized functional connectivity prior to Aβ deposition in two models, the TG2576 and the PDAPP mice, which transitioned to hyposynchronized functional connectivity later in life as Aβ deposition became more prominent [204]. Treatment with the murine form of bapineuzimab, the 3D6 antibody, prevented the fMRI by both delaying and preventing future hyposynchrony. Lastly, in a fibril-seeded hTau.P301L mouse model, the primary resting-state networks were unaffected, leading the authors to posit that tau aggregates do not alter functional connectivity [205].

At the present time, given the limited number of studies, it is difficult to draw firm conclusions on the utility of fMRI in AD mouse models. There are a number of intriguing findings, including the biphasic functional connectivity response, that should be explored in depth in a broad range of AD models.

### 5.7. Manganese-Enhanced MRI (MEMRI)

Activity-dependent MEMRI has been used sparingly in AD mouse models. The APP mouse exhibited decreased axonal transport from the hippocampus to septal nuclei and amygdala; additional transport decrements were identified in the visual system [206]. In APPSwInd (Swedish and Indiana mutations) mice, there was decreased axonal transport with increasing age, and these decrements were exacerbated in aged APPSwInd mice [207]. Tg2576 axonal transport rates were decreased with increasing Aβ load [208]. An interesting study in APPxPS1-Ki mice found that axonal transport was unable to confirm the known neuronal hypoactivity that was ascribed to uncontrolled BBB leakage [209]. The authors urged caution in MEMRI use when the BBB might be compromised in AD mouse models.

### 5.8. Susceptibility-Weighted Imaging (SWI) and Quantitative Susceptibility Mapping (QSM)

The recent advent of SWI to map venous compartments and brain iron content has led to a small number of studies. SWI in APP/PS1 mice was able to map hypointense areas in the hippocampus consistent with iron deposition that, upon immunohistopathology, demonstrated iron accumulation in microglia, particularly those associated with Aβ deposits [210]. In Tg2576 mice, T2/T2* was able to visualize the progressive deposition of Aβ plaques, which are known to be associated with iron [211]. QSM was able to monitor the progressive Aβ deposition in Tg-SwDI mice from 6 to 12 months of age [212].

SWI and QSM have been utilized in only a handful of studies but demonstrate utility for localizing in vivo Aβ deposits. In the future, more studies should investigate quantitative SWI or QSM in mapping brain iron content, which is known to increase with age.

In summary, there is a wealth of neuroimaging studies in a broad range of mouse models that can provide insights into human AD. With increasing age, many of these mouse models showed volume loss (regional and global), reduced vascular perfusion with altered vascular topology, and early consistent decrements in many brain metabolites, most notably those associated with neurons and glia. Diffusion MRI shows considerable promise in mapping gray and white matter changes, but additional studies are needed to clearly identify if this imaging modality has biomarker potential. Other imaging studies utilizing fMRI, MEMRI and SWI are underrepresented, but have potential to answer specific physiological questions in AD mouse models.

## 6. Comments on Rigor and Reproducibility in AD Preclinical Neuroimaging

As can be inferred from the studies above, there is limited concordance between studies, with some reporting alterations and others finding no change with advancing age. Even when changes are reported, some groups found opposite findings with increases or decreases. For example, in the MRS studies, broadly, there was equal representation in citations with increases, decreases or no change. A number of factors contribute to these divergent findings. Firstly, the use of neuroimaging (even at high MR field strengths) is difficult to undertake in a small mouse brain, and also reflects the complexity of AD progression in these mice models. A surprising number of studies did not report the sex of the mouse, which is critical, as there are known sex differences (see Table 2 and Table 3) [213]. Additionally, there is known genetic drift in many transgenic models, and diligent efforts are necessary to monitor and maintain the genomic validity of the transgenic AD mouse models of interest [214]. Variations in MRI sequence parameters, as well as the resolution at which data are acquired, can differ considerably between investigators and imaging sites. Similarly, at present, there is no consensus in the AD preclinical imaging field on the best practices for data pre-processing and post-processing routines. Methods for the generation of MR-derived parametric maps also differ. There is also considerable breadth in how regions of interests (ROIs) are derived, ranging from manual drawing and the use of Allen Brain Atlas, to MRI-specific mouse brain atlases. Additionally, histopathological correlations with the derived MRI results are critical to provide a comprehensive understanding of the temporal evolution of AD progression in these mouse models and their relationships to the neuroimaging findings. In summary, there are many factors that can influence the reported MR outcomes and increased efforts need to be directed to provide increased methodological transparency in the published literature. The generation of shareable AD mouse model databases focused on multi-model MR imaging that are made available to the larger AD research community would also advance the field. One nascent effort is the AD Knowledge Portal, sponsored by NIA (https://adknowledgeportal.synapse.org/, accessed on 30 December 2021).

## 7. Summary

The preclinical neuroimaging of AD mouse models provides a basis to examine brain-wide alterations that may or may not mirror clinical findings. AD is a multifactorial disease process that encompasses all cell types in the brain, from endothelial cells to neurons and glial cells. The synergy or noncooperation between these cell types impacts vessels, microglia and astrocytes (inflammation), which ultimately leads to neuronal loss and synaptic decline. Neuroimaging can facilitate the monitoring of brain cell types and their functions.

In the 5xFAD model, which is known to have accelerated AD-like pathological features, MRI is beginning to grow, particularly as new transgenic models are being developed. MEMRI studies for the most part have reported increased neuronal activity, which contrasts with either no change or decreased metabolism (^18^F-FDG-PET). Overt volumetric changes were not reported in the 5xFAD mice, but may become apparent in aged mice (>18 months). Likewise, brain metabolites were decreased in general, although age- and sex-specific differences have been reported. No diffusion MRI studies have been reported. Broadly, as 5xFAD mice age, brain-wide and hippocampal-specific alterations appear that generally reflect age-related pathology.

In the 3xTg-AD mouse model, brain-wide and regional volumetric decrements along with increased ventricular volumes are evident with increasing age. Decreased metabolisms (via ^18^F-FDG-PET) were reported with some regional specificity. Diffusion MRI changes were reported in the hippocampus and other brain regions consistent with increasing neuropathology in the 3xTg-AD mouse brain. Therefore, preclinical neuroimaging assessments can identify increasing pathological perturbations in AD mouse models.

In this review, we have highlighted that both clinical and preclinical neuroimaging studies can assist in the diagnosis of AD, as well as illuminate underlying pathology and the progression of disease. In a modest effort to link neuroimaging modalities, Table 4 represents a targeted list of studies in which there is concordance between human and animal imaging findings.

The reader should note that there exist studies in which the proposed concordance is not as persuasive (as we have noted above), but in general, human findings are mirrored in AD mouse models and vice versa. It is important to note that both MR and PET imaging modalities can readily identify frank Aβ deposits and the resultant neurobiological consequences in late-stage AD in human patients and in animal models. However, a significant gap in AD research still exists, in that most neuroimaging methods cannot reliably predict MCI onset nor the initial progression to AD. The emerging use of CSF or blood-borne biomarkers will facilitate diagnostic accuracy in AD diagnosis [227]. Additionally, artificial intelligence or machine learning methods are emerging as tools to predict AD onset and progression [228].

In sum, neuroimaging (MRI, PET) is an important and necessary tool for AD diagnosis in patients, and contributes greatly to our understanding of the pathophysiology in AD mouse models, particularly when the disease is present. The research avenues we have highlighted in this review are important, but equally critical is the need for neuroimaging to be able to predict AD onset. We look forward to novel advances from neuroimaging in being able to identify, monitor and assess treatments for AD, so as to alleviate the burden of AD in patients and their families.

## Figures and Tables

**Figure 1 biomedicines-10-00305-f001:**
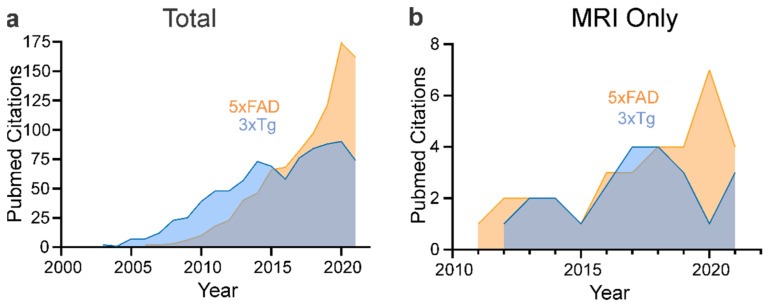
Pubmed citations for 5xFAD and 3xTg-AD mouse models. (**a**) The total number of Pubmed citations for the two most popular AD mouse models, illustrates the rapid rise in reports utilizing the 5xFAD compared to the 3xTg-AD mouse models of AD. (**b**) When the Pubmed search criteria were further refined to only 5xFAD or 3xTg-AD with MRI, the number of citations dropped precipitously. Again, the 5xFAD mouse model has an increased utilization in neuroimaging studies (Pubmed search “5xFAD” or “3xTg” and then combined with “magnetic resonance imaging”).

**Figure 2 biomedicines-10-00305-f002:**
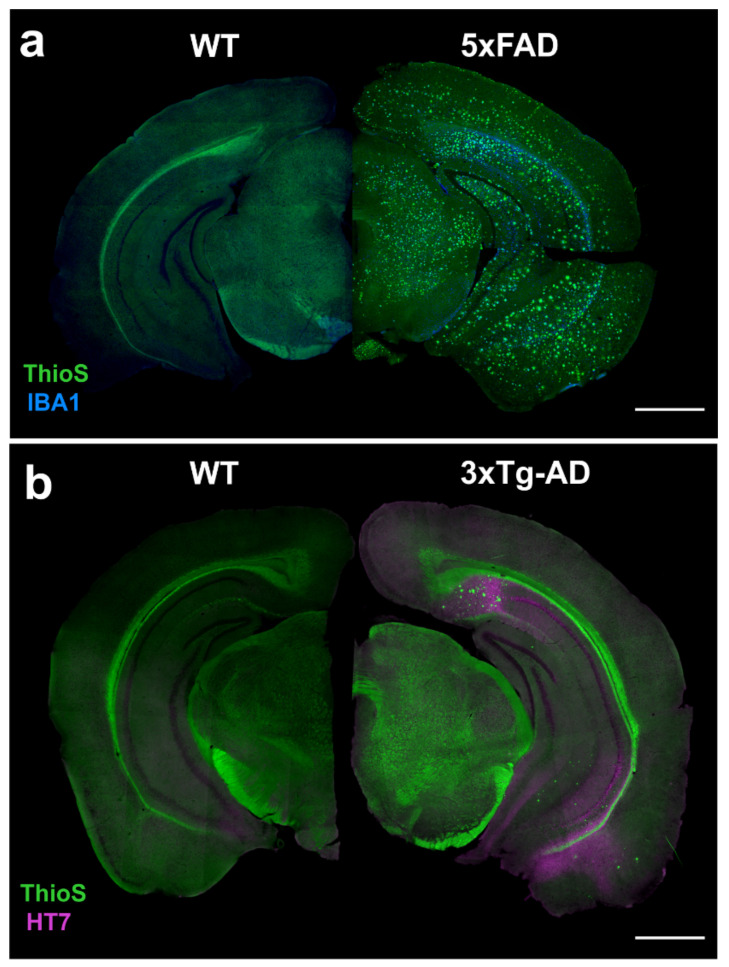
WT and transgenic female mice histological staining. (**a**) Staining in 5xFAD females at 18 months of age demonstrated a high load of β-amyloid plaques (Thioflavin-S) throughout the brain accompanied by elevated microglial activation (IBA-1) that was not observed in WT mice. (**b**) Staining for human Tau (HT7) as well as β-amyloid plaques (Thioflavin-S) in 18-month-old 3xTg-AD females revealed a more focal staining pattern compared to the 5xFAD model at the same age. Note the relative absence of β-amyloid plaques in the 3xTg-AD mice. Scale bar = 1 mm. (Images courtesy of K. Green, UCI MODEL-AD Consortium).

**Figure 3 biomedicines-10-00305-f003:**
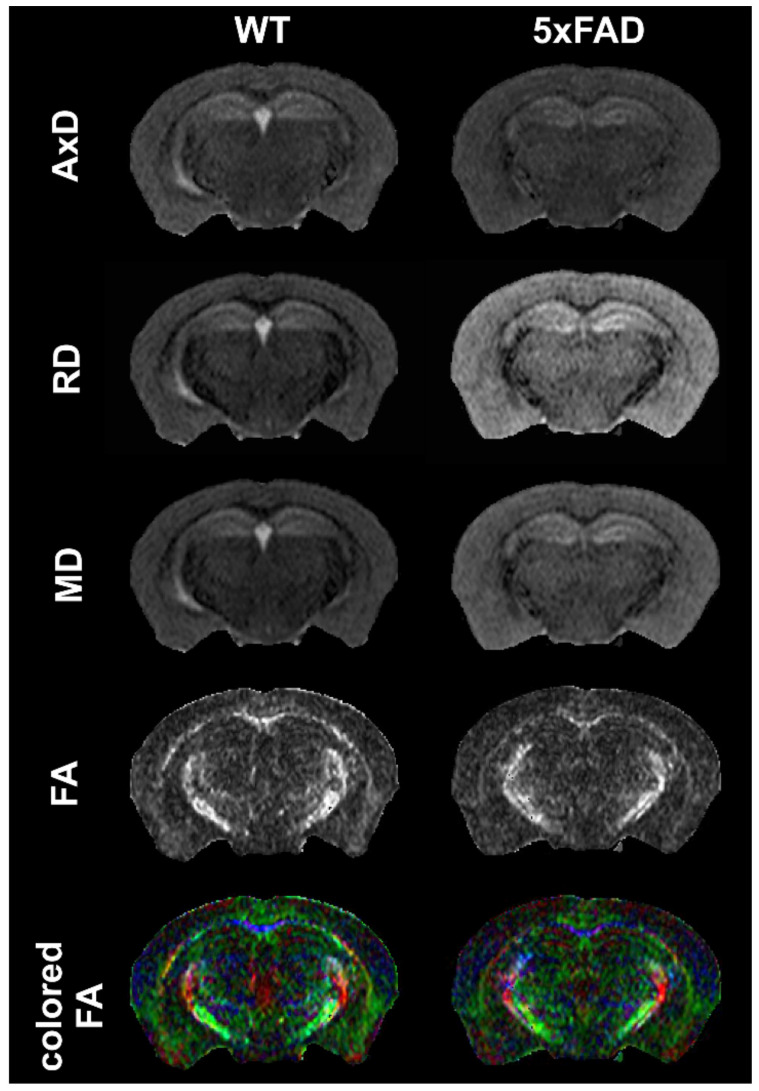
Diffusion magnetic resonance imaging (dMRI) in 12-month-old female mice. Diffusion tensor imaging (DTI) parametric maps for axial diffusivity (AxD), radial diffusivity (RD), mean diffusivity (MD), fractional anisotropy (FA) and colored FA in females WT and 5xFAD at 12 months of age. Exemplar images at the level of the dorsal hippocampus illustrate changes in the corpus callosum and within the hippocampus.

**Table 1 biomedicines-10-00305-t001:** Magnetic resonance imaging contrasts available for preclinical AD research.

MR Sequence	Information Provided
T2-weighted imaging (T2WI)	Regional volumes and multi-echo regional tissue relaxation for iron and water content.
Diffusion tensor imaging (DTI)	Tissue microstructure. Regional axial, radial and mean diffusivity and fractional anisotropy. Multishell DTI enables alternate reconstruction schemes such as NODDI for neurite density (NDI), dispersion (ODI) and isotropic water (ISOVF) indices.
Susceptibility-weighted imaging (SWI)	Iron associated with amyloid β deposition, iron content and extravascular blood; useful in cerebral amyloid angiopathy. Newer QSM methods allow for quantification.
Perfusion-weighted imaging (PWI)	Cerebrovascular function, cerebral blood volumes and flow.
Functional MRI (fMRI)	Resting state (rsfMRI) for brain-wide connectivity and task evoked functional MRI for task-specific connectivity.
Spectroscopy (MRS)	Brain metabolites

Abbreviations: NODDI—neurite orientation dispersion and density imaging. QSM—quantitative susceptibility mapping. rsfMRI—resting state functional MRI.

**Table 2 biomedicines-10-00305-t002:** Summary of 5xFAD mice imaging.

References	Imaging Age (Months) *	Sex	Imaging Modality	Magnet (T)	In Vivo, Ex Vivo, In Vitro	Key Findings
Mlynarik et al., 2012 [63]	8, 9, 10, 16	NS	MRS	14.1	in vivo	Increased Myo, decreased NAA and GABA at 9 months in dorsal hippocampus
Aytan et al., 2013 [64]	8	F	MRS	14	in vitro	Decreased NAA, GABA and Glu, increased Myo and Gln in the motor cortex
Spencer et al., 2013 [65]	11	M, F	MRI: T1, T2	4.7	in vivo	Lower T1 and T2 values in 5xFAD mice, T1 more sensitive to change
Girard et al., 2013 [66]	2, 4, 6	NS	MRI: T2 RARE	7	in vivo	No differences in volumes (whole brain, forebrain, cerebral cortex, ventricles, frontal cortex, hippocampus, striatum, olfactory bulbs)
Rojas et al., 2013 [67]	10–16	NS	PET		in vivo	Detection of Aβ with ^11^C-PiB and ^18^F-Florbetapir, increased ^18^F-FDG uptake in 5xFAD compared to WT
Girard et al., 2014 [68]	2, 4, 6	M, F	MRI: T2 RARE	7	in vivo	No differences in volumes (whole brain, forebrain, cerebral cortex, ventricles, frontal cortex, hippocampus, striatum, olfactory bulbs)
Macdonald et al., 2014 [69]	2, 5, 13	M	PET-CT, MRI	3	in vivo	Reduced ^18^F-FDG uptake and 10% decrease in hippocampal volume at 13 months, no other volume differences
Tang et al., 2016 [70]	1, 2, 3, 5	M	MRI: T1, MEMRI	7	in vivo	Increased signal intensity in brain regions involved in spatial cognition
Aytan et al., 2016 [71]	3	F	MRS	14	in vitro	Decreased taurine, NAA, GABA and Glu in the hippocampus
Mirzaei et al., 2016 [72]	6	F	PET		in vivo	Uptake of ^11^C-PBR28 is increased in 5xFAD mice
Spencer et al., 2017 [73]	2.5, 5	M, F	MRI: T1	4.7	in vivo	T1 is not a sensitive measure to detect disease onset or progression at early stages
DeBay et al., 2017 [74]	5	M	PET-CT		in vivo	Decreased ^18^F-FDG uptake in 5xFAD vs. WT
Kesler et al., 2018 [75]	6	M	MRI: fMRI, T2RARE, DTI	7, 9.4	in vivo, ex vivo	Structural networks exhibited higher path lengths in vivo and ex vivo 5xFAD vs. WT
Lee M et al., 2018 [76]	10	M	MRI: T2, PET	9.4	in vivo	^18^F-FPEB shows mGluR5 is decreased in hippocampus and striatum of 5xFAD mice
Son et al., 2018 [77]	9.5	F	PET		in vivo	Decreased ^18^F-FDG uptake in 5xFAD vs. WT
Oh et al., 2018 [78]	5.5	M	PET-CT		in vivo	Used ^11^C-FC119S to quantify Aβ in 5xFAD brain
Nie et al., 2019 [79]	1, 2, 3, 5	M	MRI: MEMRI	7	in vivo	Increased neuronal activity in hippocampus and amygdala at 1, 2, and 5 months
Son et al., 2020 [80]	10	F	PET, microCT		in vivo	Binding of DR2 tracer (^18^F-Fallypride) is decreased in 5xFAD mice, no differences/effects seen with mGluR5 tracer (^18^F-FPEB)
Oh et al., 2020 [81]	9	NS	PET-CT		in vivo	Decreased ^18^F-FPEB uptake in 5xFAD mice
Frost et al., 2020 [82]	14	NS	PET-MRI	7	in vivo	Increased uptake of ^18^F-Florbetapir in cortex, hippocampus and thalamus in 5xFAD mice
Franke et al., 2020 [83]	7, 12	M	PET-MRI	1	in vivo	Detection of cerebral hypometabolism and increased plaque load before the onset of severe memory deficits
Cho et al., 2020 [84]	NS	NS	PET-CT		in vivo	Testing of new ^64^Cu tracers to detect Aβ
Rejc et al., 2021 [85]	longitudinal from 2 to 12	F	PET-CT		in vivo	Increased ^18^F-Florbetaben uptake in cortex and hippocampus starting at 5 months Possible use of ^11^C-BChE as biomarker
Kim et al., 2021 [86]	8–9.5	M, F	MRI: MEMRI, SWI	9.4	in vivo	Manganese improves SNR, but SWI alone is sufficient to detect amyloid plaques
Chang et al., 2021 [87]	6	M	MRI: microvascular MRI	7	in vivo	Microvascular damage in the cortex of 5xFAD mice
Tataryn et al., 2021 [88]	7, 12	F	MRI: DSC-MRI, PET-CT	7	In vivo	No significant differences in whole brain glucose uptake between WT and 5xFAD
Oblak et al., 2021 [89]	4, 12	M, F	PET-MRI	3	In vivo	Increased ^18^F-FDG retention in cortex of 5xFAD females at 12 months Using ^18^F-Florbetapir, detection of Aβ at 4 months and significant increase by 12 months

NS—not specified. * all ages are in months, if reported in weeks then rounded to the nearest month.

**Table 3 biomedicines-10-00305-t003:** Summary of 3xTg-AD mice imaging.

References	Imaging Age (Months) *	Sex	Imaging Modality	Magnet (T)	In Vivo, Ex Vivo	Key Findings
Algarzae et al., 2012 [102]	12, 18	NS	MRI: T2	7	in vivo	Cortical atrophy and increased ventricular volumes at 18 months
Ishihara et al., 2013 [103]	6	NS	MRI: T1 w Gd	1.5	in vivo	No differences in BBB permeability
Kastyak-Ibrahim et al., 2013 [104]	11, 13, 15, 17	NS	MRI: T2 and DTI	7	in vivo, ex vivo	No detectable white matter changes (volumes, DTI metrics or myelin staining)
Sancheti et al., 2013 [105]	7, 13	NS	PET-CT		in vivo	Decreased ^18^F-FDG uptake in 3xTg-AD mice compared to WT
Sancheti et al., 2014 [106]	7	M	MRS, PET		ex vivo	50% increased influx of ^13^C in 3xTg-AD mice compared to controls, no differences in ^18^F-FDG uptake in the hippocampus and motor and somatosensory cortex of 3xTg-AD mice compared to WT
Hohsfield et al., 2014 [107]	7, 14, 20	M	MRI: T2*FLASH, SWI	9.4	ex vivo	No microbleeds found, increased ventricle size in 3xTg-AD mice at 14 and 20 months
Wu Z et al., 2015 [108]	22	M, F	MRI: T2	9.4	in vivo	No volume differences in cortex or hippocampus between 3xTg-AD and WT mice
Ye M et al., 2016a [109]	6	NS	PET		in vivo	Decreased ^18^F-FDG uptake in 3xTg-AD mice compared to WT in hippocampus and prefrontal cortex
Ye M et al., 2016b [110]	6	NS	PET		in vivo	Decreased ^18^F-FDG uptake in 3xTg-AD mice compared to WT in diencephalon
Baek et al., 2016 [111]	6	M	PET		in vivo	Decreased ^18^F-FDG uptake in 3xTg-AD mice compared to WT
Snow et al., 2017 [112]	12, 14	M, F	MRI: T2RARE, EPI DTI	7	in vivo	Changes of DTI metrics in hippocampus but not in cortex
Dudeffant et al., 2017 [113]	from 11 to 24	M	MRI: 3D GRE w Gd	7	in vivo, ex vivo	Amyloid plaques could not be detected with MRI in 3xTg-AD mice
Montoliu-Gaya et al., 2018 [114]	6	F	MRI: T2RARE, MRS	7	in vivo	No significant difference in whole brain volume, increased alanine in cortex and hippocampus of 3xTg-AD mice
Kong et al., 2018 [115]	1.5, 2, 3, 4, 5, 6	M, F	MRI: 3D Flash T1WI, MEMRI	7	in vivo	Decreased brain regions volume
Adlimoghaddam et al., 2019 [116]	11	M	PET-MRI	7	in vivo	Decreased ^18^F-FDG uptake in the bilateral piriform area and insular cortex of 3xTg-AD mice compared to WT, but no differences in the whole brain
Chiquita et al., 2019 [117]	4, 8, 12, 16	M	MRI: T2, 2D DCE-FLASH, PET	9.4	in vivo	Decreased hippocampal volume at all ages, decreased BBB permeability index at 16 months, decreased taurine levels in hippocampus, no difference in ^11^C-PiB and ^11^C-PK11195 uptake
Manno et al., 2019 [118]	2	F	MRI: T2RARE, DTI, rsfMRI	7	in vivo	4-fold increase in ventricle volume, decreased hippocampal interhemispheric connectivity, increased cortical FA but decreased RD
Rollins et al., 2019 [119]	2, 4, 6	M, F	MRI: MEMRI	7	in vivo	Decreased brain regions volume
Guëll-Bosch et al., 2020 [120]	5, 7, 9, 12	F	MRI: T2RARE, T2MSME, EPI, MRS	7	in vivo	Decreased brain volume, increased Aβ, increased inflammation in hippocampus and cortex
Falangola et al., 2021 [121]	2, 8, 15	NS	MRI: dMRI	7	in vivo	Changes in DTI metrics in 3xTg-AD mice compared to WT at 2 and 8 months
Stojakovic et al., 2021 [122]	16	M, F	PET-CT		in vivo	Decreased ^18^F-FDG uptake in males and females 3xTg-AD mice compared to WT
Chen et al., 2021 [123]	5, 8, 11	NS	PET-CT		in vivo	Increased ^11^C-PiB in 8- and 11-month-old 3xTg-AD mice, increased uptake of the HDAC tracer ^18^F-TFAHA at 8 and 11 months

NS—not specified. * all ages are in months, if reported in weeks then rounded to the nearest month.

**Table 4 biomedicines-10-00305-t004:** Concordance between human and preclinical neuroimaging findings.

Imaging Modality	Generalized Findings	Human Studies	Mouse Studies
MRI–Structural	Volumetric decreases Brain atrophy in human studies Less robust findings in mouse AD models	Schroeter et al., 2009 [39], Jobson et al., 2021 [40]	Girard et al., 2014 [68], Mcdonald et al., 2014 [69], Hohsfield et al., 2014 [107], Guëll-Bosch et al., 2020 [120], Lau et al., 2008 [130]
MRI-dMRI	Increased FA Decreased MD, AxD, RD Reduced connectivity	Nir et al., 2013 [46], Chen et al., 2020 [50]	Manno et al., 2019 [118], Falangola et al., 2021 [121], Qin et al., 2013 [182], Shu et al., 2013 [183]
MRI–multishell dMRI	Increased ODI Decreased NDI Increased intracellular volume fraction (ICVF)	Wen et al., 2019 [47], Fu et al., 2020 [48]	Colgan et al., 2016 [189], Colon-Perez et al., 2019 [191]
rs-fMRI	Decreased connectivity (temporal lobe) Increased path lengths; increased disconnectivity	Schwindt et al., 2009 [55]	Kesler et al., 2018 [75], Manno et al., 2019 [118], Shah et al., 2016 [204]
MRI–MRS	Decreased NAA Decreased NAA/Cr Decreased Glu/Gln (Glx) Increased Myo	Jessen et al., 2009 [215], Modrego and Fayed 2012 [216], Foy et al., 2011 [217], Walecki et al., 2011 [218]	Mlynarik et al., 2012 [63], Guëll-Bosch et al., 2020 [120], Oberg et al., 2008 [148]
PET-FDG	Reduced metabolism	Silverman et al., 2001 [219], Levin et al., 2021 [220]	Son et al., 2018 [77], Franke et al., 2020 [83], Adlimoghaddam et al., 2019 [116]
PET-Aβ	Increased uptake	Sintini et al., 2020 [24], Panegyres et al., 2009 [25]	Rojas et al., 2013 [67], Frost et al., 2020 [82], Chen et al., 2021 [123]
PET–Tau	Increased uptake with advancing AD Tau labeling recapitulates Braak staging	Cho et al., 2020 [27], Johnson et al., 2016 [28], Vogel et al., 2020 [221]	Brendel et al., 2016 [222], Sahara et al., 2017 [188]
PET- glial	Increased microglial binding associated with atrophy Increased astrocyte binding	Femminella et al., 2016 [223], Nicastro et al., 2020 [224], Kumar et al., 2021 [225]	Mirzaei et al., 2016 [72], Rodriguez-Vieitez et al., 2015 [226]

## Data Availability

Not applicable.

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
