# Peer review of "Neuroimaging of Mouse Models of Alzheimer’s Disease"

_biomedicines, 2022, doi:10.3390/biomedicines10020305_

Round 1
Reviewer 1 Report
Authors provide an excellent and insightful review. Article is well written and really informative.
I would suggest to add this reference to complete the review of literature.
PMID: 34987086
This article is interesting for the reader since the article is about diagnosis of AD. This field need to have article like this to provide new insights and raise new diagnosis avenues. The reader of this article should have a clear overview of what we do with MRI at this time and why AD is a complex disease. This article has all the qualities required to be published. I provided a PMID to include in the manuscript which talk about AD and its complexity. I found it relevant. The scientific characteristics of this paper is excellent and relevant. They observe different parameters of AD using MRI and also discuss the limitations of the model.Author Response
Response to Reviewer 1 Comments
Authors provide an excellent and insightful review. Article is well written and really informative.
Point 1: I would suggest to add this reference to complete the review of literature.
PMID: 34987086
Response 1: We thank the reviewer for their comments. Even though the suggested review is very interesting (Targeting Systemic Innate Immune Cells as a Therapeutic Avenue for Alzheimer’s Disease; Pons and Rivet 2022), we do not believe it fits within the scope of our review. The suggested manuscript tackles potential therapeutic targets through innate immunity but does not touch on how neuroimaging would track these targets. Since our focus is only on neuroimaging and not on potential treatments, we have elected not to include this reference.
Reviewer 2 Report
This is a well compiled, exhaustive, and well articulated review encompassing our contemporary understanding of the preclinical neuroimaging of AD mouse models. The review is multifaceted, meticulously compiled, and well organized. The following minor suggestions may increase the merit of the manuscript.
- Synaptic loss is considered the most accurate correlate of memory deficits and cognitive decline in Alzheimer's disease. A multitude of recent studies have highlighted the application of a wide range of neuroimaging techniques and methods to identify derangements in synaptic density and synaptic loss (examples include PET-based imaging of synaptic markers as a correlate of synaptic density). The authors could add an excerpt addressing this facet in their review.
- Astrogliosis and reactive microgliosis are a cardinal pathological hallmark of early Alzheimer's disease pathology. A plethora of studies have applied contemporary imaging techniques to characterize this facet of neuropathology in a wide spectrum of mouse models of Alzheimer's disease. The authors could address this aspect in their review.
- Presynaptic dystrophic neurites and alterations in postsynaptic dendritic spine density and morphology are a characteristic hallmark of Alzheimer's disease and the 5xFAD mouse model also exhibits this pathological hallmark. The authors could reflect on this facet of Alzheimer's disease pathology in their review.
Author Response
Response to Reviewer 2 Comments
This is a well compiled, exhaustive, and well articulated review encompassing our contemporary understanding of the preclinical neuroimaging of AD mouse models. The review is multifaceted, meticulously compiled, and well organized. The following minor suggestions may increase the merit of the manuscript.
Point 1: Synaptic loss is considered the most accurate correlate of memory deficits and cognitive decline in Alzheimer's disease. A multitude of recent studies have highlighted the application of a wide range of neuroimaging techniques and methods to identify derangements in synaptic density and synaptic loss (examples include PET-based imaging of synaptic markers as a correlate of synaptic density). The authors could add an excerpt addressing this facet in their review.
Response 1: Thank you for pointing this out. We have added a several sentences in Section 2.1 to discuss PET imaging for synaptic loss through SV2A labeling.
Revision, Page 3, line 128
“Synaptic loss has also been evaluated in patients with mild cognitive impairment and AD using ligands for synaptic vesicle protein 2A (SV2A). 11C-UCB-J and 18F-UCB-J both highlighted hippocampal synaptic loss and correlated with cognitive decline.”
Point 2: Astrogliosis and reactive microgliosis are a cardinal pathological hallmark of early Alzheimer's disease pathology. A plethora of studies have applied contemporary imaging techniques to characterize this facet of neuropathology in a wide spectrum of mouse models of Alzheimer's disease. The authors could address this aspect in their review.
Response 2: We thank the reviewer for this suggestion as well. In Section 2.1 right after the PET imaging of activated microglia, we have added a sentence to note PET imaging of astrogliosis, albeit there is not much information.
Revision, Page 3 line 125
“Other ligands targeting astrogliosis such as 11C-Deuterium-L-Deprenyl (11C-DED) are being tested as predictive factors to detect AD prior to symptomology. A recent in-depth review explores glial PET imaging.”
Point 3: Presynaptic dystrophic neurites and alterations in postsynaptic dendritic spine density and morphology are a characteristic hallmark of Alzheimer's disease and the 5xFAD mouse model also exhibits this pathological hallmark. The authors could reflect on this facet of Alzheimer's disease pathology in their review.
Response 3: We are grateful for this comment. We have included synaptic loss and dystrophic neurites as a characteristic hallmark of AD in patients as well as in the 5xFAD mice (in Sections 1 and 4.1).
Revision, Page 1 line 42
“While the exact biological mechanisms of pathogenesis are unclear in LOAD, cellular hallmarks include the extracellular accumulation of beta-amyloid (Aβ) peptides into senile plaques, the intracellular accumulation of hyperphosphorylated tau into neurofibrillary tangles (NFTs), glial (astrocytic and microglial) responses, neuronal and synaptic loss.”
Revision, Page 7 line 263
“Presynaptic dystrophic neurites are also detected in areas surrounding amyloid plaques at 5-6 months.”
Reviewer 3 Report
I write this review as a clinician-scientist who has been involved in the diagnosis, therapeutics and lab-based investigations of neurodegenerative diseases (including AD) for 32 years. As the authors know, the diagnosis and therapy of AD have substantial scientific, medical, economic and social/political implications. The review article by Jullienne, et al, must be judged in those contexts, not just as a review of scientific data.
Jullienne, et al, have provided a very comprehensive (exhaustive, really) review of imaging abnormalities in the two most popular mouse models of familial AD (FAD), which, as they point out early in their paper, affect 5% or less of the AD population. What Jullienne, et al, do not provide, but which I feel they are very capable of providing, is some kind of overview (a Table would do this nicely) of how the imaging of the brains in transgenic (TG) mouse models correlate (or not) with findings in humans with sporadic (and maybe the few with familial) AD.
As the authors also likely know, much current AD research addresses the question of brain insulin resistance in AD. Disease-altering treatments may emerge with therapies that increase brain insulin sensitivity in AD, esp since amyloid removal immuno-treatments appear to have substantial side effects and unclear efficacies.
In conclusion, I support publication of the current paper without changes, but would suggest that the authors include two additional items:
- a comparison (Table ?) of imaging modalities in FAD mouse models and humans with AD
- a brief discussion of brain insulin resistance in FAD mouse models and human AD, with particular reference to any PET glucose studies or other imaging studies. The authors are also free to suggest future additional imaging related to brain insulin resistance.
Author Response
Response to Reviewer 3 Comments
I write this review as a clinician-scientist who has been involved in the diagnosis, therapeutics and lab-based investigations of neurodegenerative diseases (including AD) for 32 years. As the authors know, the diagnosis and therapy of AD have substantial scientific, medical, economic and social/political implications. The review article by Jullienne, et al, must be judged in those contexts, not just as a review of scientific data.
Jullienne, et al, have provided a very comprehensive (exhaustive, really) review of imaging abnormalities in the two most popular mouse models of familial AD (FAD), which, as they point out early in their paper, affect 5% or less of the AD population. What Jullienne, et al, do not provide, but which I feel they are very capable of providing, is some kind of overview (a Table would do this nicely) of how the imaging of the brains in transgenic (TG) mouse models correlate (or not) with findings in humans with sporadic (and maybe the few with familial) AD.
As the authors also likely know, much current AD research addresses the question of brain insulin resistance in AD. Disease-altering treatments may emerge with therapies that increase brain insulin sensitivity in AD, esp since amyloid removal immuno-treatments appear to have substantial side effects and unclear efficacies.
In conclusion, I support publication of the current paper without changes, but would suggest that the authors include two additional items:
Point 1: a comparison (Table ?) of imaging modalities in FAD mouse models and humans with AD
Response 1: We thank the reviewer for this constructive suggestion. Table 4 was now added to section 7 to list studies in which there is concordance between human and animal imaging findings:
Table 4. Concordance between human and preclinical neuroimaging findings
|
Imaging Modality |
Generalized Findings |
Human Studies |
Mouse Studies |
|
MRI – Structural |
Volumetric decreases, Brain atrophy in human studies, Less robust findings in mouse AD models |
Schroeter et al 2009 [39], Jobson et al 2021 [40]
|
Girard et al 2014 [68], Mcdonald et al 2014 [69], Hohsfield et al 2014 [107], Guëll-Bosch et al 2020 [120], Lau et al 2008 [130] |
|
MRI - dMRI |
Increased FA, Decreased MD, AxD, RD, Reduced connectivity |
Nir et al 2013 [46], Chen et al 2020 [50] |
Manno et al 2019 [118], Falangola et al 2021 [121], Qin et al 2013 [182], Shu et al 2013 [183] |
|
MRI – multishell dMRI |
Increased ODI, Decreased NDI, Increased intracellular volume fraction (ICVF) |
Wen et al 2019 [47], Fu et al 2020 [48] |
Colgan et al 2016 [189], Colon-Perez et al 2019 [191] |
|
rs-fMRI |
Decreased connectivity (temporal lobe), Increased path lengths; increased disconnectivity |
Schwindt et al 2009 [55] |
Kesler et al 2018 [75], Manno et al 2019 [118], Shah et al 2016 [204] |
|
MRI – MRS |
Decreased NAA, Decreased NAA/Cr, Decreased Glu/Gln (Glx), Increased Myo |
Jessen et al 2009 [215], Modrego and Fayed 2012 [216], Foy et al 2011 [217], Walecki et al 2011 [218] |
Mlynarik et al 2012 [63], Guëll-Bosch et al 2020 [120], Oberg et al 2008 [148] |
|
PET - FDG |
Reduced metabolism |
Silverman et al 2001 [219], Levin et al 2021 [220] |
Son et al 2018 [77], Franke et al 2020 [83], Adlimoghaddam et al 2019 [116] |
|
PET - Aβ |
Increased uptake |
Sintini et al 2020 [24], Panegyres et al 2009 [25] |
Rojas et al 2013 [67], Frost et al 2020 [82], Chen et al 2021 [123] |
|
PET – Tau |
Increased uptake with advancing AD, Tau labeling recapitulates Braak staging |
Cho et al 2020 [27], Johnson et al 2016 [28], Vogel et al 2020 [221] |
Brendel et al 2016 [222], Sahara et al 2017 [188] |
|
PET- glial |
Increased microglial binding associated with atrophy, Increased astrocyte binding |
Femminella et al 2016 [223], Nicastro et al 2020 [224], Kumar et al 2021 [225] |
Mirzaei et al 2016 [72], Rodriguez-Vieitez et al 2015 [226] |
Point 2: a brief discussion of brain insulin resistance in FAD mouse models and human AD, with particular reference to any PET glucose studies or other imaging studies. The authors are also free to suggest future additional imaging related to brain insulin resistance.
Response 2: Several sentences were added in Section 4.1.6 to discuss the effect of insulin levels and insulin resistance as it relates to 18F-FDG uptake:
Revision, Page 12 line 398
“This contrasts with the increased 18F-FDG retention in cortical regions of 12-month-old female 5xFAD mice compared to WT [88], possibly due to the fact that mice in the Tataryn study were fasted prior to PET imaging. A clinical study demonstrated that increased plasma glucose levels, but not plasma insulin or insulin resistance levels, could explain the decreased 18F-FDG uptake [94]. As shown by Fueger et al., the biodistribution of 18F-FDG is influenced by fasting, body temperature and the type of anesthesia used. These parameters should be standardized to allow for proper comparisons between future studies [95]. Additional studies are clearly needed to clarify these divergent metabolic studies.”